# Peak Slope Ratio of the Recruitment Curves Compared to Muscle Evoked Potentials to Optimize Standing Configurations with Percutaneous Epidural Stimulation after Spinal Cord Injury

**DOI:** 10.3390/jcm13051344

**Published:** 2024-02-27

**Authors:** Ahmad M. Alazzam, William B. Ballance, Andrew C. Smith, Enrico Rejc, Kenneth A. Weber, Robert Trainer, Ashraf S. Gorgey

**Affiliations:** 1Spinal Cord Injury and Disorders Center, Richmond VA Medical Center, Richmond, VA 23249, USA; ahmadalazzam6@gmail.com (A.M.A.); wbb2ballance@gmail.com (W.B.B.); 2Physical Therapy Program, Department of Physical Medicine and Rehabilitation, University of Colorado School of Medicine, Aurora, CO 80045, USA; andrew.c2.smith@cuanschutz.edu; 3Department of Medicine, University of Udine, 33100 Udine, Italy; enrico.rejc@uniud.it; 4Department of Anesthesiology, Perioperative and Pain Medicine, Stanford University School of Medicine, Palo Alto, CA 94304, USA; kenweber@stanford.edu; 5Department of Physical Medicine and Rehabilitation, Virginia Commonwealth University, Richmond, VA 23284, USA; robert.trainer@va.gov; 6Physical Medicine and Rehabilitation, Richmond VA Medical Center, Richmond, VA 23249, USA

**Keywords:** percutaneous spinal cord epidural stimulation, recruitments curves, slope ratio, spinal cord injury, rehabilitation

## Abstract

**Background:** Percutaneous spinal cord epidural stimulation (pSCES) has effectively restored varying levels of motor control in persons with motor complete spinal cord injury (SCI). Studying and standardizing the pSCES configurations may yield specific motor improvements. Previously, reliance on the amplitude of the SCES-evoked potentials (EPs) was used to determine the correct stimulation configurations. **Methods:** We, hereby, retrospectively examined the effects of wide and narrow-field configurations on establishing the motor recruitment curves of motor units of three different agonist–antagonist muscle groups. Magnetic resonance imaging was also used to individualize SCI participants (*n* = 4) according to their lesion characteristics. The slope of the recruitment curves using a six-degree polynomial function was calculated to derive the slope ratio for the agonist–antagonist muscle groups responsible for standing. **Results:** Axial damage ratios of the spinal cord ranged from 0.80 to 0.92, indicating at least some level of supraspinal connectivity for all participants. Despite the close range of these ratios, standing motor performance was enhanced using different stimulation configurations in the four persons with SCI. A slope ratio of ≥1 was considered for the recommended configurations necessary to achieve standing. The retrospectively identified configurations using the supine slope ratio of the recruitment curves of the motor units agreed with that visually inspected muscle EPs amplitude of the extensor relative to the flexor muscles in two of the four participants. Two participants managed to advance the selected configurations into independent standing performance after using tonic stimulation. The other two participants required different levels of assistance to attain standing performance. **Conclusions**: The findings suggest that the peak slope ratio of the muscle agonists–antagonists recruitment curves may potentially identify the pSCES configurations necessary to achieve standing in persons with SCI.

## 1. Introduction

Restoration of motor recovery to achieve standing, stepping, or overground locomotion was not previously feasible in persons with spinal cord injury (SCI). The introduction of spinal cord epidural stimulation (SCES) began to alter these medical beliefs and facilitated motor recovery after SCI. Over the last decade, several reports have demonstrated the use of SCES to spatiotemporally activate the lumbosacral neural circuitries and restore motor recovery [1,2,3,4,5,6,7,8]. The use of SCES facilitates the integration of supraspinal, interspinal, and propriospinal afferent signals to generate motor behavior. These studies rely on distinct stimulation patterns and specific electrode sites to achieve complex tasks such as locomotion [9]. This is even true in persons with clinically defined motor complete SCI (i.e., American Spinal Injury Association Impairment Scale (AIS) A and B) [1]. Since clinically motor complete spinal cord injuries are often not anatomically complete [10], previously dormant neural fibers traversing the lesion site may be able to activate neuronal pools distally in the lumbosacral cord in the presence of neuromodulation [11,12]. Therefore, it is possible to assume that the extent of the spinal cord lesion may influence motor outcomes after SCI.

Magnetic resonance imaging (MRI) measures of the spinal cord lesion can help quantify the extent of the spinal cord lesion, and the amount of spared neural fibers as quantified by MRI may help explain the ability to achieve simple voluntary lower extremity movements during SCES. Previously, our research group showed that overground locomotion is related to the spinal cord axial damage ratio as measured by MRI [13,14]. Another study reported a relationship between the standing motor behavior using SCES and MRI measures of spared neural fibers [14]. A case series reported a high correlation with MRI-measured spinal cord lesion length and lower extremity motor score improvement using transcutaneous spinal cord stimulation [15]. Other groups clearly indicated that MRI detects the extent of neurological deficit or recovery after SCI [16,17]. Therefore, it is assumed that injury characteristics may influence motor behavior in persons with SCI.

Previous work organized and developed sequential algorithms to optimize motor behavior in persons with SCI [18,19]. When a 16-electrode array is surgically implanted in individuals with SCI with the goal of promoting lower limb motor recovery [20], intraoperative evoked potentials are elicited by subsets of electrode contacts with the goal of optimizing electrode placement [9,21]. After post-operative recovery, spinal mapping provides critical information about the spatial relationship between stimulation sites and SCES evoked potentials of specific muscles and results in an individualized map of motor pool activation. Similarly, testing a range of stimulation frequencies is helpful to estimate the type of activation pattern (i.e., tonic vs. bursting activation pattern) that will be promoted while attempting to facilitate a motor task. This information is necessary for selecting the optimal subset of stimulation parameters that will be tested and refined under an experimental setting to facilitate a given motor task [21]. In addition to stimulation frequencies, muscle activation thresholds and amplitude–response curves are evaluated to determine whether key proximal and distal muscles are preferentially activated by rostral and caudal contacts, respectively, and whether there is consistency in side-specific (i.e., right vs. left side musculature) activation [22]. The resultant motor pattern can be better described in terms of the adjusted anodal–cathodal configuration as well as the stimulation parameters in a spatiotemporal manner to the neural circuitries. The relationships between characteristics of evoked potentials (EP) in lower limb muscles and intensities of SCES (i.e., recruitment curves) can provide important insights on the effects of the stimulation parameters as well as on the neurophysiological structures impacted by spinal stimulation [23]. This is highly important because SCES for sensorimotor recovery is a burgeoning field and there is a crucial need to determine the optimal stimulation configurations to achieve beneficial outcomes.

During the recovery of standing, the individualized map of motor pool activation is used to adjust the position of cathodes in order to target extensor muscle groups while limiting the activation of flexor muscles [21]. Similarly, the range of stimulation frequencies eliciting a tonic activation pattern is subsequently tested during standing to find the optimum frequency promoting an overall continuous standing activation pattern [24]. Therefore, detailed analysis of recruitment curves elicited in a supine position by SCES provides additional neurophysiological insights. For example, characteristics of the slope of recruitment curves provide important information about the properties of afferents projecting to specific motoneuron pools and the rate of type Ia afferent recruitment, among others [25,26]. In addition, quantifying lesion characteristics and estimates of spared neural fibers may be important for the overall understanding of the effects of SCES on enabling specific voluntary motor behavior.

Percutaneous spinal cord epidural stimulation (pSCES) has been used to restore motor recovery in persons with SCI [27,28,29]. The term “percutaneous” refers to the specific surgical technique implemented in the placement of the leads in the epidural space. One of the two participants showed restoration of locomotor-like activity that facilitated overground ambulation [27]. Additionally, we have further explored the potential of using interleaved configurations after implantation of pSCES to restore overground mobility [28]. These reports successfully highlight the distinct motor behavior of distinct muscle groups after manipulating stimulation configurations and parameters, using a percutaneous single lead in persons with complete SCI [29]. A step-by-step description of how altering stimulation configurations of pSCES may influence motor patterns in relation to MRI-measured lesion characteristics is still lacking. In this report, we retrospectively evaluated the recruitment curves that eventually lead to restoring standing control in four persons with SCI. Evaluation of the recruitment curves was completed in response to different stimulation configurations (wide vs. narrow-field configurations), different pulse duration (250, 500, and 1000 µs), and after controlling for injury characteristics in persons with SCI. In current human studies, unlike animal work, we do not have access to specific automated algorithms that can rapidly explore the best SCES stimulation parameters [30,31]. Therefore, we hypothesized that using the peak slope ratio of the agonist–antagonist recruitment curves of different muscle groups may serve as a step towards identifying the appropriate pSCES configurations necessary to achieve standing.

## 2. Materials and Methods

### 2.1. Human Subjects

Four men with clinically motor complete traumatic SCI (C8; 6 years post-injury [ID#: 0881], T11; 9 years post-injury [ID#: 0882], T6; 12 years post-injury [ID#: 0883], T4; 24 years post-injury [ID#: 0884]) participated in a trial approved by the Richmond Veteran Affairs Medical Center (Table 1). All participants provided written informed consent about the purpose of the study, electrode implantation, stimulation, and subsequent publication of findings, which was approved by the Institutional Review Board at the Richmond VA Medical Center. All methods were performed in accordance with the relevant guidelines and regulations. The trial was registered at clinical trials.gov on 3 April 2021 with a registration ID# NCT04782947.

### 2.2. Timeline of the Study

The four participants enrolled in an approximately 12-month study that examined the effect of resistance training (RT) and pSCES (REST-SCI) on restoration of overground locomotion in persons with SCI. The focus of the first 24 weeks was directed towards enhancing standing performance, and the following 24 weeks were directed towards restoring overground locomotion. During the trial, participants practiced (3× per week) using exoskeletal assisted walking (EAW; for approx. 1 h), and this was followed by a standing re-training program for an additional 1 h. Participants were randomized into either pSCES + EAW + RT (ID#: 0881, 0882 and 0884) or delayed pSCES + EAW + noRT (ID#: 0883) groups. The delayed pSCES underwent implantation 6 months after enrollment in the trial. Participants underwent measurements at baseline (prior training), post-measurement 1 (end of the first 24 weeks), and post-measurement 2 (end of the second 24 weeks).

The inclusion and exclusion criteria were previously listed in detail [27]. Briefly, persons with a traumatic, motor complete spinal cord injury (SCI) with a neurological level of injury below C5 and age between 18 to 60 years were included. The upper limit of age was set at 60 years to protect against development of cardiovascular issues that could prevent participating in strenuous physical activity during the course of the trial. All participants underwent International Standards for Neurological Classification of SCI (ISNCSCI) [32] examination for neurological level and function, and only participants with an AIS A or B injury (indicating a motor deficit below the level of injury) were included. The inclusion requirement for AIS A and B was in place to facilitate the detection of any subsequent voluntary control below the motor injury level. Finally, participants had to be 24 months post-SCI to be considered for the trial.

### 2.3. Magnetic Resonance Imaging

Before the clinical trial, participants underwent magnetic resonance imaging with the following clinical scan (slice thickness: 3 mm, TR: 9350, TE: 102; flip angle: 150) for pre-screening purposes. MRI was conducted in order to identify the site of injury and determine the SCI lesion severity. Fifteen axial spinal cord images were acquired using a General Electric 3.0 T Signa Discovery MR750 scanner (GE Healthcare, Chicago, IL, USA) and a two-dimensional fast spin echo sequence (slice thickness = 5 mm, slice spacing = 10 mm, field of view = 200 × 200 mm^2^, matrix size = 320 × 224, repetition time = 9015 ms, echo time = 104 ms). Fifteen sagittal images of the spinal cord were acquired using a two-dimensional fast relaxation fast spin echo sequence (slice thickness = 3 mm, slice spacing = 4.5 mm, field-of-view = 360 × 360 mm^2^, matrix size = 512 × 512, repetition time = 2939 ms, echo time = 95 ms). All MRI measurements were completed by an experienced researcher who was blinded to the study design and the outcomes of each participant. The spinal cord lesion hyperintensity was manually segmented throughout each axial slice, followed by the surrounding spinal cord with manual tracing using OsiriX MD version 14.0 (Pixmeo SARL, Bernex, Switzerland) to measure the axial damage ratio biomarker. Axial damage ratio was calculated for each slice as Area_lesion_/Area_cord_, then the maximum ratio was used for each participant. In previous work, this method was demonstrated to have high inter- and intra-rater reliability [13,33,34]. Lesion length was measured using the midsagittal T2-weighted image as the most cranial to most caudal ends of the lesion hyperintensity [17,35,36,37,38]. This method has been demonstrated to have high inter- and intra-rater reliability based on previous work from a co-author’s laboratory [34].

### 2.4. Implantation of Percutaneous SCES

This is a two-step process where temporary implantation precedes permanent implantation of pSCES [27]. The pSCES system (Intellis Epidural Stimulator, Medtronic, Minneapolis, MN, USA) was used to electrically stimulate the lumbosacral enlargement. Details on the surgical implantation of the temporary percutaneous leads have been previously reported [27].

Briefly, adequate informed consent was obtained, and the patient was taken to the procedure room and placed in the prone position on the fluoroscopy table. A time out was held for patient verification. The skin was prepped and draped in the normal sterile fashion. A nurse certified in sedation established IV access, placed standard ASA monitors (Philips Medical Systems, BG Eindhoven, The Netherlands) including noninvasive blood pressure every 5 min, pulse oximetry, continuous EKG, and end tidal CO_2_ from a nasal cannula. The patient received 2 g of Ancef prior to the procedure. The patient was adequately sedated throughout the procedure to allow them to interact with the surgeon without undue anxiety or distress. The L3 vertebral body was squared and centered using Anteroposterior (AP) fluoroscopy. One percent lidocaine was used for local skin infiltration of the tract and the lamina of L3. Next two 14-gauge Medtronic 6-inch needles were guided to the interlaminar space between L2 and L3 vertebral bodies at an angle of approximately 30 degrees. From this point, a loss of resistance was obtained by use of a glass syringe with normal saline and air. After access to the epidural space was obtained with antero-posterior and lateral fluoroscopy views. From this point, the stylette was removed and 2 Medtronic leads were placed through each of the 2 needles from the right of the midline and left of the midline and into the epidural space under live AP fluoroscopy. The top of the electrode was at the top of the T11, and the 8 contacts of each lead spanned the spine to the L1 vertebral body covering the area of the conus medullaris [27].

The Medtronic representative interrogated the lead to ensure normal epidural impedance while the lateral view of fluoroscopy confirmed posterior epidural space placement. The location of the leads was verified by visually inspecting contraction of the paralyzed lower extremity muscles. The needles were then removed under live fluoroscopy to maintain the leads at the same position; final X-rays were taken after Dermabond and Steri-Strips were applied to anchor the lead to the skin. The area was covered with OpSite and gauze with tape on top of it to provide more cutaneous wound stability. The Medtronic Bluetooth receiver was placed in a pouch on the lateral low back and confirmed communication with the interrogator [27].

Approximately 2–4 weeks following temporary implantation, two 8-electrode arrays of Vectris lead were implanted in an operating room. Phase 1 of the permanent implantation is identical to the temporary implantation trial as previously described [27], but with anesthesia provided by an anesthesiologist. An IV line was established for sedation, and standard ASA monitors were placed including noninvasive blood pressure every 5 min, pulse oximetry, continuous EKG, and end tidal CO_2_ from a nasal cannula. Antibiotics (typically Cefazolin 2–3 g or Clindamycin 600–900 mg) were used at the time of implantation. Two separate 5-inch 14-gauge epidural needles were placed using X-ray guidance and loss of resistance technique to access the epidural space. The leads were navigated in the epidural space and correct configurations (i.e., stimulation parameters) necessary to induce electrical activation of the targeted muscles were re-tested. A cut down was made in the midline to expose the leads and remove them from the needle under live fluoroscopy. A second incision was made to house the battery, and after the leads were anchored to ligament and/or fascia with non-absorbable sutures, they were tunneled to the pocket and connected to the battery. Impedances were checked after hemostasis and irrigation was applied. Following hemostasis, the wound was closed in 2–3 layers; derma-bond, occlusive dressing, and tape were placed over the wound. An abdominal binder was provided for the participant’s comfort [27].

Prescription for pain medicine was only for 3 days as this seems to be the most painful part as the single incision heals. Recovery after implant was complete at the 7–10 days mark when bandages were removed. Permanent implantation was followed with 3–4 weeks of instruction not to perform strenuous physical activities without immobilization.

### 2.5. Spinal Segmental Mapping

Following permanent implantation, participants were scheduled to perform the process of spinal segmental mapping [27,29]. Spinal mapping was carried out to identify optimal cathodal–anodal electrode arrangement and stimulation parameters (frequency, amplitude, and pulse durations) to enable multiple functions and movements without inducing unwanted activity [39,40]. Spinal mapping was conducted daily after permanent implantation (2 weeks), as well after the first 24 weeks of the study in an interim mapping phase (4 weeks). Every effort was considered to configure the correct combinations of cathodes and anodes per specific muscle group and joints as well as the correct stimulation parameters for the frequency of the pulses (2–40 Hz) and (250–1000 μs) for the pulse duration. We aimed to use the minimum amount of current (1–10 mA) necessary to evoke muscle contraction and to monitor the increase or decrease during the trial. Participants 0881 and 0882 underwent spinal mapping to develop the recruitment curves in weeks 26–29 of the study. Participants 0084 and 0883 underwent spinal mapping in weeks 5 and 33, respectively.

### 2.6. Developing Recruitment Curves

From supine position, 12 EMG sensors were attached to the following muscles ((left and right vastus medialis (VM), rectus femoris (RF), tibialis anterior (TA), medial hamstring (HS), medial gastrocnemius (MG), and gluteus medius (GM)) after shaving and carefully cleaning the skin. The participant was then asked to make three major movements such wiggling the big toe, dorsiflexion of the ankle joint, or moving the entire leg into flexion followed by extension. Multiple electrode combinations with the correct stimulation parameters were tested to identify the most optimal SCES configurations for standing. SCES-evoked recruitment curves from 5 standardized SCES configurations were established based on EMG peak-to-peak amplitude of each muscle group.

Recruitment curves were collected at two wide-field configurations (caudal cathodes vs. caudal anodes) and three narrow-field configurations (central, caudal, and rostral cathodes) at pulse durations of 250, 500, or 1000 µs (Figure 1D). From these curves, optimal stimulation configurations that could yield tonic extensor activity for standing were determined. These optimal stimulation configurations were further examined in a standing frame or standing with a standard walker to establish the standing configurations. Participants were then trained on how to use the pSCES controller to activate paralyzed lower extremity muscles.

### 2.7. EMG Data Analysis

Surface EMG signals were recorded from left and right muscles. All EMGs were collected at a 2000 Hz sampling rate using LabChart 8.1.21 (Windows, AD Instruments, Sydney, Australia). EMGs shown in figures were rectified and band pass filtered (10–990 Hz).

Briefly, the EMG of 10 subsequent pulses were rectified and then root mean square was applied to provide the area under the curve of the EP (V) in response to every stimulating intensity (mA; 1–10 mA) [26,41]. The EP (V) was then adjusted to the maximum EMG amplitude (Vmax) for each individual muscle group. The recruitment curve was then established based on the adjusted EP (V/Vmax) at each corresponding intensity (1–10 mA). After, the pSCES-EP was established at each corresponding intensity. A typical recruitment curve was fitted as shown in Figure 1F.

Using MATLAB (MathWorks 2023a), a particle swarm optimization was used to find optimal parameters for a 4-parameter sigmoid fitted to each recruitment curve. The swarm size and maximum iterations were set to 500. The recruitment curve was then visually inspected and then it was fitted in a sigmoidal function. Fits were visually inspected to ensure a sigmoidal nature, using both visual inspection and cut-off of R^2^ ≥ 0.8. Curves that did not pass visual inspection were excluded. A 6th order polynomial fit was then applied to remaining curves to measure the peak slope of every recruitment curve. The maximum slope of the fit was representative of the recruitment curve inflection slope (Figure 1F). The slope ratios between agonistic–antagonistic muscle groups were then calculated: VM/HS, GM/RF, and MG/TA. A Appendix A highlighting a step-by-step process and how the slope of the recruitment curve was fitted (Appendix A).

## 3. Results

### 3.1. Magnetic Resonance Imaging (MRI) Biomarkers of Spinal Cord Damage

Figure 2 displays sagittal T2-weighted image, sagittal lesion length, axial T2-weighted image, and axial damage ratio (Figure 2A–D). Participants 0881 and 0882 had lesion lengths of 32.0 and 23.8 mm, respectively. Additionally, participant 0881 presented with the highest axial damage ratio (0.92), while both 0882 and 0884 had similar axial damage ratios (0.81 and 0.80, respectively) (Table 2). Participant 0883 had the highest estimated lesion length of 88.1 mm; lesion volume and axial damage ratio were unable to be assessed for participant 0883 due to image artifact (Figure 2C). Participant 0884 (Figure 2D) had the lowest lesion length of 15.5 mm, while lesion volume (mm^3^) was unable to be assessed due to image artifact (Table 2), The lesion characteristics indicated that three out of the four participants were considered discomplete SCI as indicated by the axial damage ratio. The two participants who were classified as AIS A and B had comparable axial damage ratios.

### 3.2. pSCES-Evoked Supine Recruitment Curves

Analyses of the supine recruitment curves indicated that rostral and caudal cathodal placements of the pSCES resulted in a varying degree of recruitment in both proximal and distal muscles. Overall, wide-field configurations delivered with caudal or rostral cathodes promoted higher evoked potentials (EP) of the extensor muscles (e.g., VM, GM, and MG). The pSCES-evoked pattern of recruitment of hip, knee, and ankle extensors and flexors using the narrow-field configurations displayed higher variability across participants. Figure 3 demonstrated the placement of the pSCES leads and the achieved standing configurations. The standing configurations in Figure 3 were based on visual inspection of the pSCES-knee extensor EP relative to the hamstrings muscle groups. These configurations (2 Hz) were then tested in standing position using tonic stimulation (>10 Hz) to achieve the standing configurations (Figure 3).

### 3.3. Wide versus Narrow-Field Configurations

Figure 4 shows the SCES-EP collected for each standardized configuration for the knee extensors and flexors for 0882 participant. Similarly, Figure 5 and Figure 6 show the SCES-EP collected for the hip and ankle extensors and flexors.

In participant 0881, we observed distinguishable patterns of muscle recruitment among the central, caudal, and rostral narrow-field configurations. The magnitude of the VM muscle EP were the greatest when delivering the central narrow-field configuration at 500 µs and 1000 µs (Appendix A), while the caudal and rostral narrow-field configurations elicited a higher amplitude of the HS EP across different pulse durations (Appendix A). Central and caudal narrow-field configurations resulted in a greater magnitude of the GM EP compared to the RF (Appendix A), while the rostral narrow-field configuration elicited an overall minimal response of both the GM and RF m (Appendix A).

In participant 0882, the greatest magnitude of the RF and GM potentials was evoked using the wide-field configuration with rostral anodes and caudal cathodes at different pulse durations (Figure 4A and Figure 5A). In contrast, the magnitude of the VM and RF EP was greater when the wide-field configuration with rostral cathodes and caudal anodes was delivered at each pulse duration (Figure 4B and Figure 5B).

In addition to the varying degree of muscle response between wide- and narrow-field configurations, recruitment curves exhibited a unilateral leg muscle activation. For instance, in participant 0883, when delivering the caudal narrow-field configuration at 500 and 1000 µs elicited a greater VM magnitude only for the right leg (Appendix A), while very minimal response was observed for the left leg. However, using the wide-field configuration with caudal cathodes at 500 µs resulted in greater VM EPs at higher stimulation intensities (Appendix A). Additionally, all the five configurations at each pulse duration resulted in a greater magnitude of the MG compared to the TA response (Appendix A).

In participant 0884, the wide-field configuration with caudal cathodes promoted a greater HS magnitude at 250 and 500 µs, however at 1000 µs as the stimulation intensity increased the magnitude of the VM was greater (Appendix A). Minimal response from the VM was recorded at 250 µs and 1000 µs when cathodes were placed rostrally (Appendix A); however, the caudal narrow-field configuration at 250, 500, and 1000 µs resulted in a greater HS response (Appendix A).

### 3.4. Peak Slope Ratio of SCES-Evoked Recruitment Curves

The recommended configurations based on the peak slope ratio (Table 3) were provided compared to the standing configuration (Table 4) that were highlighted in Figure 3. Peak slope ratios based on the wide-field and narrow-field configurations are provided in Table 3. Retrospective analysis of the peak slope of SCES-evoked supine recruitment curves provided additional insight regarding the responsiveness of individual extensor vs. flexor muscle pairs to the five stimulation configurations.

A ratio greater than 1 indicated that pSCES resulted in greater extensor EP to flexor EP. Table 4 highlighted the agreement in configurations between the pSCES EP, primarily VM relative to the HS and the peak slope ratio of three pairs of different agonist–antagonist muscle groups.

For participant 0881, the peak slope ratio between the left VM and HS at 500 µs for the wide-field configuration with caudal cathodes and the central narrow-field configuration both had a ratio greater than 1. Across the five standardized configurations, 70% of the peak slope ratios were less than 1 and were not determined clinically suitable following visual assessment of the curves for the VM and HS. A total of 15 peak slope ratios between the GM and RF (50%) and 11 peak slope ratios between the MG and TA (37%) for both wide-field configurations, central narrow-field, and caudal narrow-field were greater than 1. The narrow-field configuration with rostral cathodes did not present peak slope ratios greater than 1 and clinically suitable for any of the muscle pairs tested (Table 3).

In participant 0882, 20% of the total peak slope ratio considered for analysis were greater than 1, while 47% resulted in peak slope ratios less than 1 for the VM and HS Five out of the six peak slope ratios calculated for the left and right VM and HS m. across pulse durations were greater than 1 for the wide-field configuration with rostral cathodes and 1 peak slope ratio for the left leg at 500 µs was greater than 1 using the rostral narrow-field configuration.

Peak slope ratios between the GM and RF resulted in a total of 17 (57%) total ratios greater than 1 and were determined clinically suitable for both wide-field configurations, central, and caudal narrow-field configurations (Table 4). However, we observed a total of six (20%) peak slope ratios from the wide-field configuration with rostral cathodes and the caudal narrow-field configuration that were less than 1. In addition, a total ofeight8 (27%) peak slope ratios between the MG and TA were greater than 1 across the five standardized configurations and clinically suitable, while 33% had ratios less than 1 (Figure 7).

For participant 0883, the peak slope ratio between the left VM and HS at 500 µs was greater than 1 for the wide-field configuration with caudal cathodes and also the left VM and HS at 1000 µs for the central narrow-field configuration. However, 43% of the peak slope ratios across the five configurations tested were less than 1. We did not observe peak slope ratios greater than 1 and clinically suitable for the MG and TA; however, a total of five peak slopes were greater than 1 but were not suitable following visual assessment of the recruitment curves (Appendix A).

Moreover, we did not observe peak slope ratios greater than 1 and were clinically suitable for the VM and HS, while a total of 12 peak slope ratios were less than 1 for the wide-field configuration with caudal cathodes and the central narrow-field configuration. Additionally, a total of six peak slope ratios (33%) were greater than 1 for the MG and TA for the wide-field configuration with caudal cathodes and the caudal narrow-field configuration (Table 3).

### 3.5. Standing Performance

The pSCES lead placements and selected standing configurations were shown in Figure 3. Independent standing ability was assessed by whether participants needed external assistance from study staff at the trunk, hips, or knees to maintain a standing position in parallel bars or walker.

The standing configuration for participant 0881 was established using the wide-field configuration with caudal cathodes and the central narrow-field configuration with caudal cathodes (both delivered at 750 µs; 15 Hz), and both were interleaved constituting the third configuration, which was delivered at 700 µs; 15 Hz (Figure 3). Three SCES configurations, all with caudal cathodes and rostral anodes were delivered simultaneously to enable standing (Figure 3A). Participant 0881 achieved standing without external assistance at the trunk but required external assistance at the hips and knees (Figure 8A).

Figure 9 displays representative electromyograms (EMGs) of lower extremity muscles of participant 0882, while in a standing frame (Figure 9). Both wide-field configurations and the narrow-field configuration with rostral cathodes were re-tested in a standing frame or standing with a standard walker to establish the standing configurations (Figure 9). The wide-field configuration with rostral cathodes and caudal anodes generally promoted a more effective EMG pattern for standing. The standing configuration for participant 0882 was established using a wide-field configuration with rostral cathodes and caudal anodes and a narrow-field configuration with rostral cathodes delivered at a frequency of 15 Hz and pulse duration of 500 µs (Figure 3). Each configuration was gradually ramped up one configuration at a time, until both configurations were set at an amplitude of 3.6mA. Once the participant was able to self-assist into an upright position, 0882 was able to maintain upright standing without hip or knee support from the research team (Figure 8B).

Further, the standing configuration for participant 0883 was established using both wide-field configurations with rostral cathodes and caudal cathodes. Two SCES configurations, both with rostral cathodes and anodes were delivered at (15 Hz; 760 µs), and caudal cathodes and anodes for the second configuration (10 Hz; 370 µs). However, participant 0883 was unable to achieve overground standing and required external assistance from study staff at the hips and knees during sit-to-stand attempts (Figure 8C).

Furthermore, participant 0884 was able to achieve standing with external assistance at the hips during a sit-to-stand transition using a walker (Figure 8D). Once the participant was in an upright position, assistance at the hips and knees was decreased gradually and the participant was able to maintain standing with the use of upper extremities. One pSCES configuration with rostral anodes and caudal cathodes was delivered at a frequency of 15Hz and a pulse duration of 700 µs (Figure 3D). However, following migration of the percutaneous leads, the pSCES configurations failed to enable standing performance.

## 4. Discussion

The major findings indicated that the peak slope ratio of the recruitment curves compared to the standard pSCES EP narrows down the configurations at different lead placements, wide vs. narrow field of stimulation, different pulse durations, and in anatomically discomplete injuries as assessed by the MRI–axial damage ratios. Three out of the four participants had wide-field configurations that elicited motor behavior that may have secured partial or independent standing behavior. One participant who failed to achieve independent standing had both wide and narrow field configurations. Our results further suggest that the MRI-measured extent of lesion and estimates of spared neural fibers may be important for understanding the responsiveness of pSCES.

### 4.1. Rationale of Conducting the Study

Standing performance was previously achieved by visual inspection of the pSCES EP between VM and HS [27,28]. The pSCES-evoked potentials were tested in a supine lying position at 2 Hz at different amplitudes (1–10 mA). This resulted in subjective selection of the configurations, which was then tested in a standing frame to achieve standing performance using tonic stimulation parameters (>10 Hz). The standing performance was then achieved through visually inspecting the EMG activities of the VM (Figure 8), receiving direct feedback from the participant on whether the configurations promote sitting vs. standing, or palpating either the VM or HS. The entire process took a considerable amount of time and elicited up to 30 recruitment curves per single muscle group without even considering the left vs. right side asymmetry, different pulse durations, and wide vs. narrow configurations. At this point, an objective tool to assess the suitable configuration for the three different pairs of agonist–antagonist muscle groups was not available. Therefore, the SCES-evoked recruitment curves were retrospectively analyzed to objectively standardize the selection procedures of the configurations necessary to achieve standing.

To move towards objectivity, the pSCES recruitment curves were retrospectively analyzed by performing visual inspection to their sigmoidal shape and then were fitted using a sigmoidal equation to determine their quality (R^2^ > 0.80). R^2^ of 0.8 was chosen as an arbitrary cut-off and based on Cohen *d* effect size [42]. The sixth degree polynomial function was then used to measure the slope of each recruitment curve. The ratio between the slopes of the agonist–antagonist muscle groups were then used to narrow down the selected curves. A slope ratio equal to or greater than 1 provided advantages to determine the pSCES configurations between wide- and narrow-field configurations, different pulse duration (250, 500, and 1000 µs), and injury characteristics with different axial damage ratio (0.8–0.92).

### 4.2. MRI and SCES Configurations

The high-resolution MRI biomarkers provide an additional understanding of the neuroanatomical characteristics of the spinal cord lesion and motor responsiveness to pSCES. Previous reports have noted a strong association with axial damage ratios and the ability to restore motor recovery in persons with SCI [14,15]. Recent data on axial damage ratios in a retrospective study involving 145 participants with SCI reported a cut-off score of 0.37 to differentiate walkers versus wheelchair users [13]. Our results are in alignment with this finding, as all four of our participants cannot walk and had axial damage ratios above this value, ranging from 0.80 to 0.92. Additionally, spared neural tissue traversing the lesion site may be important in determining the ability to volitionally stand in the presence of SCES [14], and our axial damage ratio data suggest that our participants demonstrate at least some evidence of spared neural tissue indicating a discomplete profile. Lesion length, a surrogate measure of the extent of injury, has been widely reported in the literature as having a negative association with neurological outcomes in SCI [35,43,44,45], with several studies finding a lesion length in the acute phase of less than 30mm having a favorable AIS grade improvement later on [35,45]. A previous report also showed that lesion length in the chronic phase may be related to lower extremity volitional function in the presence of transcutaneous spinal cord stimulation [15]. In the current study, our participants’ lesion length was measured in the chronic phase, and we did find that our two participants who had lengths less than 30mm (0882 and 0884) demonstrated the best volitional standing performance using optimized stimulation parameters.

### 4.3. Supine Mapping and Recruitment Curves

The level of spinal motor network responsiveness is crucial for performing motor functions in the presence of pSCES. Supine mapping allows the determination of relationships between different electrode configurations, frequency, and amplitude to understand the excitability state of these motor networks. Previously, we and others [23,24,46] relied on visual inspection of the recruitment curves and determined the appropriate configurations based on the visual inspection of the EP of the VM and HS muscle groups [27]. This was subsequently tested and modified in a standing frame to ensure that these configurations were successful in achieving standing performance. The peak slope ratios potentially provide differentiation between the sensitivity of various motor neuron pools between pairs of extensor and flexor muscle groups to different stimulation configurations.

pSCES-EP were collected in a relaxed supine position at 2 Hz and then were further tested in a standing position at different tonic frequencies. As previously noted, the pSCES configurations are position-dependent. The close proximity of the pSCES leads to the spinal cord in a supine position is attenuated by transmission of the current against circulating cerebrospinal fluid and other meninges in a standing position. This may explain why some of the configurations showing extensor features did not effectively translate into standing configurations for some participants to achieve standing. Additionally, weightbearing-related sensory information is integrated by the spinal circuitry and profoundly modulate the related motor output [1].

### 4.4. Significance of Testing SCES Configurations in Supine Position

The selection of appropriate tonic SCES parameters is critical to restore motor function after severe SCI. SCES parameters are task- and individual-specific [1,24], and the vast amount of combinations of stimulation parameters potentially available can be a significant barrier toward the broader use of SCES for functional recovery if clear guidelines on the selection of these parameters are not provided [47]. Spinal segmental mapping assessments performed in supine position [21,41] can provide useful information to identify a subset of electrode configurations to be tested for functional recovery. This is particularly important to the search of functionally effective stimulation parameters to achieve upright position (e.g., standing), especially considering the number of factors including blood pressure regulation, manual assistance provided by trainers, fatigue, and overall comfort level of the participant. The amplitude of motor unit EP elicited by 2 Hz SCES in supine position at increasing stimulation amplitudes (i.e., recruitment curves) [41], determined for different muscles and electrode configurations, is one of the outcomes considered to define the SCES electrode configurations to be tested for standing [21]. One of the key interpretations of this supine dataset is related to identifying the active electrodes that engage motor pools of primary extensor muscles (VM) while reducing the activation of flexor muscles (HS m).

Supine recruitment curves demonstrated two levels of motor excitability independent of the amplitude of the current. The first level is described as a linear activity in motor recruitment from baseline to approximately 70% or more of the maximum evoked potential (point A). The second level represents a steep rise from point A to 80% or more of the maximum evoked potential. Afterwards the recruitment curves attain a plateau regardless of the increase in the magnitude of the amplitude. These findings may suggest that different levels of motor excitability are attained during establishing the recruitment curves. Furthermore, it may confirm previous findings that highlighted the need to stimulate the spinal cord circuitries at sub-motor threshold level [26,27,28]. This would likely facilitate integration of the descending supraspinal signals with both segmental activation and ascending afferent input (i.e., subthreshold network modulation). However, we need to acknowledge that not all target segmental muscles or all configurations yield this typical pattern. Such discrepancies in motor recruitments curves may be explained by different lead placements or reflecting the status or the size of motor neuron pools. Persons with SCI commonly experienced degenerative changes of their motor neuron pool including atrophy of motor neuron size as well as shifting from fast oxidative to fast glycolytic fibers. Additionally, evidence suggests loss in slow motor units in persons with chronic SCI. This may have likely resulted in different motor recruitment patterns [27].

An alternative outcome to the maximum amplitude achieved by the recruitment curves which may provide important information for the selection of stimulation parameters is the peak slope of the recruitment curves. From a neurophysiological standpoint, this variable provides information about the maximum rate of recruitment of type Ia afferents during stimulation [25,26]. A recent publication [48] suggests that there can be muscle-specific relationships between spinal cord segments targeted by SCES and peak slope. Furthermore, larger electrode fields generally elicited higher peak slopes compared to narrower electrode fields. These observations support the view that peak slope can capture motor pool-specific behaviors induced by specific electrode configurations. However, the effects of different electrode configurations on the relationship between peak slopes of extensor vs. flexor muscle groups was not investigated, and this may represent an important outcome to translate supine mapping data into guidelines for the selection of a subset of configurations to be tested to facilitate standing.

### 4.5. Recruitment Curves Yield Different Outcomes

Our analysis indicated that only part of the recruitment curves showed a sigmoidal-like profile (i.e., typical curve), which may be considered when attempting to interpret the stimulation curves. The fitting of pSCES EP recruitment curves to a sigmoidal function elicited a range of R^2^ values, which subsequently provided us with the capability, upon visual inspection, to determine the configurations that elicited a greater slope ratio between extensor and flexor muscle groups. Examining the slope ratio between extensors and flexors provided additional guidance that configurations (wide-field vs. narrow-field) may elicit different EP response at different joints. The peak slope ratio facilitated the path on whether to consider interleaved or single configurations. Previous findings have clearly shown that interleaved configurations further enhanced the capability of a person with SCI to actively perform overground stepping [28]; however, the rationale in many cases is unclear as to why interleaved configurations are considered in a person with SCI.

## 5. Limitations

There are a few limitations that need to be acknowledged in the current study. Unlike other studies, we utilized increments of 1 mA in establishing the supine recruitment curves. Previously, increments of 0.1 mA were used up to 4 mA. This may have resulted in a loss of important information pertaining to the findings of the current report. We only examined supine recruitment curves, and it would be interesting to further explore the potential of establishing standing requirement curves on further narrowing the configurations that yield independent standing. We only studied persons with motor complete SCI and generalizing the findings to persons with incomplete SCI needs to be considered with caution. Finally, in two of our participants, pSCES leads were placed in a staggered fashion and not at equidistant level. The development of fibrous tissue at the site of injury prohibited the threading of leads in one direction compared to the other direction. The different placement of the leads may have altered the pSCES-evoked potentials and result in asymmetrical responses between the left and right sides or among the four participants [41].

## 6. Summary/Conclusions

In conclusion, peak slope ratios, determined retrospectively from recruitment curves in supine position, can be important to consider for selecting the subset of electrode configurations to facilitate standing. The peak slope ratio of the recruitment curve has the advantage over peak EP that is achieved at lower stimulation intensities, thus it can be reliably determined as stimulation intensity that may be limited by the discomfort of the participant or the effect on blood pressure regulation. Our findings demonstrate the distinctive neuroanatomical characteristics and response across SCI individuals to pSCES and the inherent variability across participants. These findings may aid researchers and clinicians in narrowing down the selection of configurations and decrease the time required to find optimal configurations for promoting standing in persons with SCI. The use of the peak slope ratio may serve as a step towards developing a machine learning interphase that automatically narrows the stimulation configurations after pSCES implantation in persons with SCI.

## Figures and Tables

**Figure 1 jcm-13-01344-f001:**
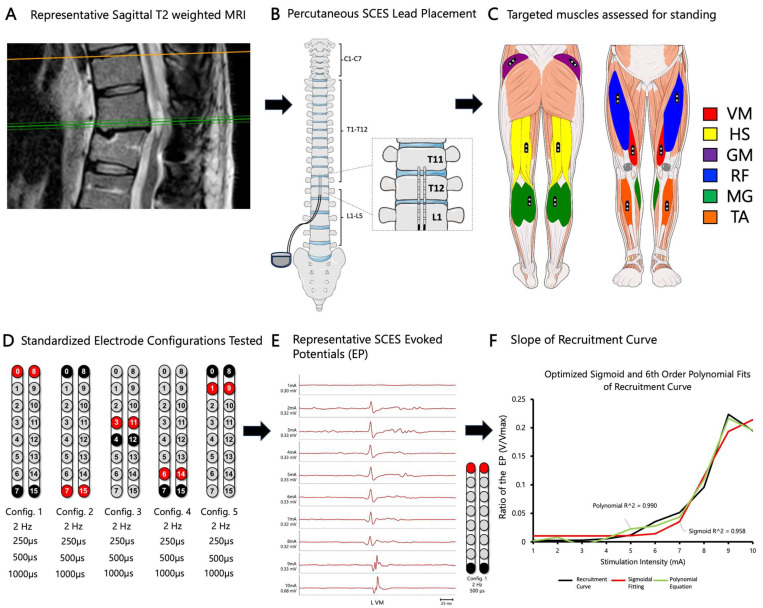
**Representative neurophysiological assessment post percutaneous spinal cord implantation.** The black arrows highlighted the sequence of events of the study according to the designed timeline. (**A**) Representative MRI sagittal T2-weighted image of lesion site (Participant ID: 0882). MRI: magnetic resonance image. (**B**) Representative anatomical illustration of the percutaneous SCES lead implantation. SCES: spinal cord epidural stimulation. (**C**) Anatomical illustration highlighting the lower extremity muscles assessed for standing using surface EMGs. EMGs: electromyograms, VM: vastus medialis, HS: hamstrings, GM: gluteus medius, RF: rectus femoris, MG: medial gastrocnemius, TA: tibialis anterior. (**D**) Standardized electrode lead configurations tested during spinal segmental mapping post implant in supine position (frequency 2 Hz, pulse width 250, 500, and 1000 µs). Configuration 1: wide-field configuration with caudal cathodes (black) and rostral anodes (red), Configuration 2: wide-field configuration with caudal anodes and rostral cathodes, Configuration 3: central narrow-field configuration, Configuration 4: caudal narrow-field configuration, Configuration 5: rostral narrow-field configuration. (**E**) Representative SCES evoked potentials collected at intensity from 1–10 mA for the left vastus medialis muscle using the wide-field configuration with caudal cathodes and rostral anodes at (2 Hz; 500 µs) (Participant ID: 0882). (**F**) Representative diagram illustrating the calculation of the maximal peak slope of the recruitment curve using a fitted sigmoid function (red) and 6th order polynomial (green). Parts of this figure were drawn by using pictures from Servier Medical Art. Servier Medical Art by Servier is licensed under a Creative Commons Attribution 3.0 Unported License (https://creativecommons.org/licenses/by/3.0/, accessed on 19 July 2023).

**Figure 2 jcm-13-01344-f002:**
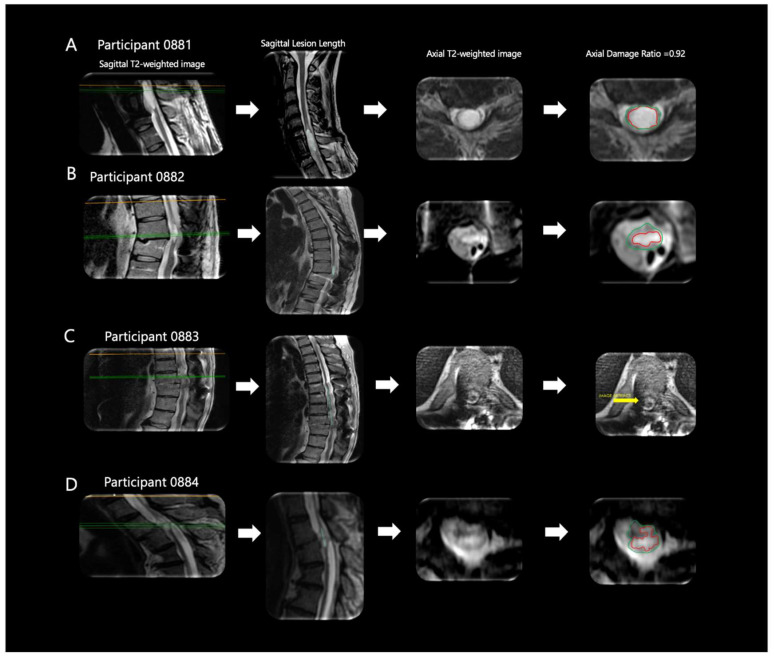
**Axial MRI Biomarkers.** Four participants’ lesion characteristics on magnetic resonance imaging (MRI). (**A**) Participant ID: 0881, (**B**) Participant ID: 0882, (**C**) Participant ID: 0883, (**D**) Participant ID: 0884. Using each participant’s sagittal T2 weighted imaging (**left column**), lesion length was quantified (**second left column**) as a measure of the extent of damage. Then, using each participant’s axial T2 weighted imaging, (**second right column**), axial damage ratios were quantified (**right column**) to provide an estimate of the amount of lesion to non-lesion tissue. The green line at each sagittal image (**left column**) denotes which axial slice was used for axial damage ratio calculation.

**Figure 3 jcm-13-01344-f003:**
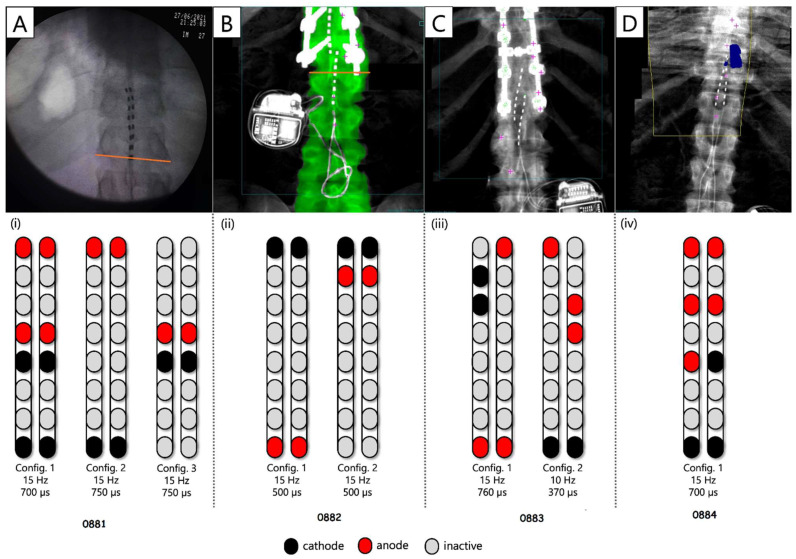
**Percutaneous SCES lead placement and optimal cathodal and anodal standing configurations without using the peak slope ratio.** Each participant underwent permanent implantation with two 8-electrode percutaneous leads to enable motor control below the level of injury. (**A**) Permanent leads for 0881 were placed at the T11-L1 vertebrae to cover the lumbosacral enlargement. (**i**) Three SCES configurations, all with caudal cathodes and rostral anodes were delivered simultaneously to enable standing. (**B**) Permanent leads for 0882 necessitated staggered positioning of the left lead covering (T12-L1) and the right lead covering (L1-L2) due to the presence of spinal hardware and excessive scar tissue. (**ii**) Two SCES configurations, both with rostral cathodes and a mix of rostral and caudal anodes were delivered simultaneously to enable standing. (**C**) Permanent leads for 0883 were staggered to cover the top of T11-L1, left lead covering (T11) and right lead covering (L1). (**iii**) Two SCES configurations, both with rostral cathodes and anodes and caudal cathodes and anodes were delivered simultaneously to assist in standing. (**D**) Permanent leads for 0884 were placed at the T10-T11 to mid-T12 vertebrae. (**iv**) One SCES configuration with rostral anodes and caudal cathodes was delivered to enable standing.

**Figure 4 jcm-13-01344-f004:**
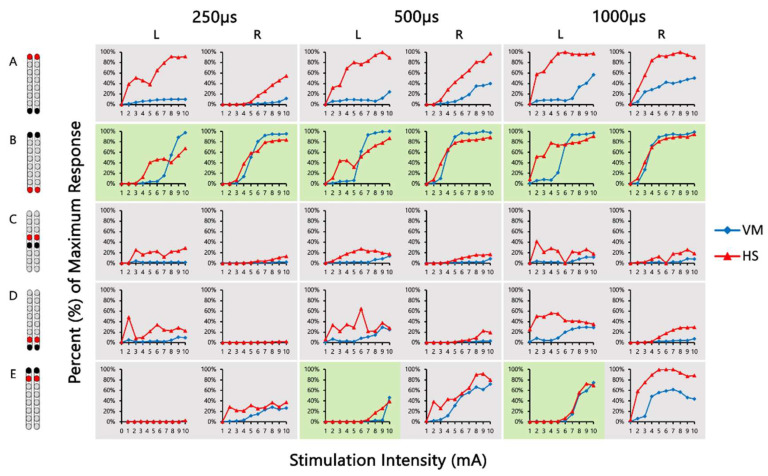
**Supine SCES-evoked recruitment curves of the knee extensor and flexor muscles for 0882.** Supine SCES-evoked recruitment curves of the left and right VM and HS muscles collected by stimulating at 2 Hz from (1–10 mA) using the 5 standardized SCES configurations each collected at pulse durations of 250, 500, and 1000 µs. (**A**) Wide-field configuration with caudal cathodes and rostral anodes. (**B**) Wide-field configuration with rostral cathodes and caudal anodes. (**C**) Central narrow-field configuration with rostral anodes and caudal cathodes. (**D**) Caudal narrow-field configuration with rostral anodes and caudal cathodes. (**E**) Rostral narrow-field configuration with rostral cathodes and caudal anodes. Recruitment curves highlighted in green indicate an overall higher extensor to flexor muscle response, while curves highlighted in gray represent an overall higher flexor to extensor muscle response. Responses were normalized to the maximum response (Vmax) of each muscle across all stimulation intensities (mA) in all configurations. Cathode and anode electrodes are shown in black and red, respectively. VM: vastus medialis; HS: hamstrings; mA: milliampere; µs: microsecond; L: left; R: right.

**Figure 5 jcm-13-01344-f005:**
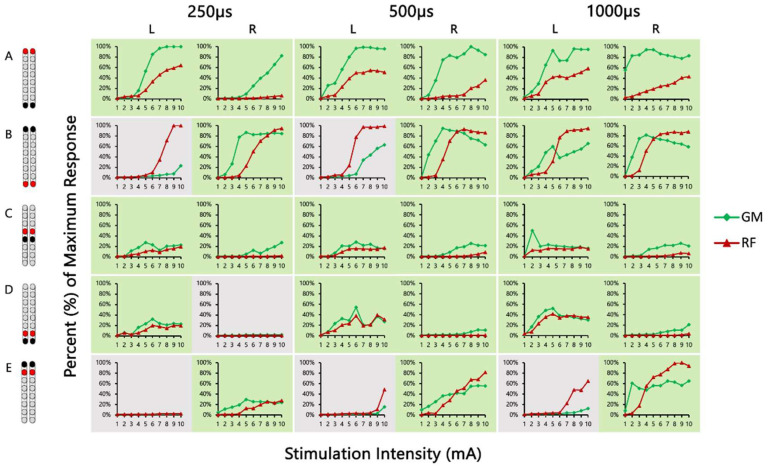
**Supine SCES-evoked recruitment curves of the hip extensor and flexor muscles for 0882.** Supine SCES-evoked recruitment curves of the left and right GM and RF muscles collected by stimulating at 2 Hz from (1–10 mA) using the 5 standardized SCES configurations each collected at pulse durations of 250, 500, and 1000 µs. (**A**) Wide-field configuration with caudal cathodes and rostral anodes. (**B**) Wide-field configuration with rostral cathodes and caudal anodes. (**C**) Central narrow-field configuration with rostral anodes and caudal cathodes. (**D**) Caudal narrow-field configuration with rostral anodes and caudal cathodes. (**E**) Rostral narrow-field configuration with rostral cathodes and caudal anodes. Recruitment curves highlighted in green indicate an overall higher extensor to flexor muscle response, while curves highlighted in gray represent an overall higher flexor to extensor muscle response. Responses were normalized to the maximum response (Vmax) of each muscle across all stimulation intensities (mA) in all configurations. Cathode and anode electrodes are shown in black and red, respectively. GM: gluteus medius; RF: rectus femoris; mA: milliampere; µs: microsecond; L: left; R: right.

**Figure 6 jcm-13-01344-f006:**
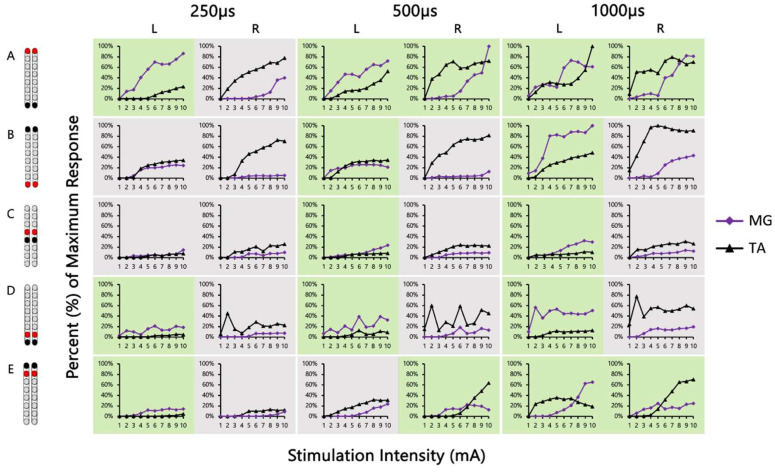
**Supine SCES-evoked recruitment curves of the ankle extensor and flexor muscles for 0882.** Supine SCES-evoked recruitment curves of the left and right MG and TA muscles collected by stimulating at 2 Hz from (1–10 mA) using the 5 standardized SCES configurations each collected at pulse durations of 250, 500, and 1000 µs. (**A**) Wide-field configuration with caudal cathodes and rostral anodes. (**B**) Wide-field configuration with rostral cathodes and caudal anodes. (**C**) Central narrow-field configuration with rostral anodes and caudal cathodes. (**D**) Caudal narrow-field configuration with rostral anodes and caudal cathodes. (**E**) Rostral narrow-field configuration with rostral cathodes and caudal anodes. Recruitment curves highlighted in green indicate an overall higher extensor to flexor muscle response, while curves highlighted in gray represent an overall higher flexor to extensor muscle response. Responses were normalized to the maximum response (Vmax) of each muscle across all stimulation intensities (mA) in all configurations. Cathode and anode electrodes are shown in black and red, respectively. MG: medial gastrocnemius; TA: tibialis anterior; mA: milliampere; µs: microsecond; L: left; R: right.

**Figure 7 jcm-13-01344-f007:**
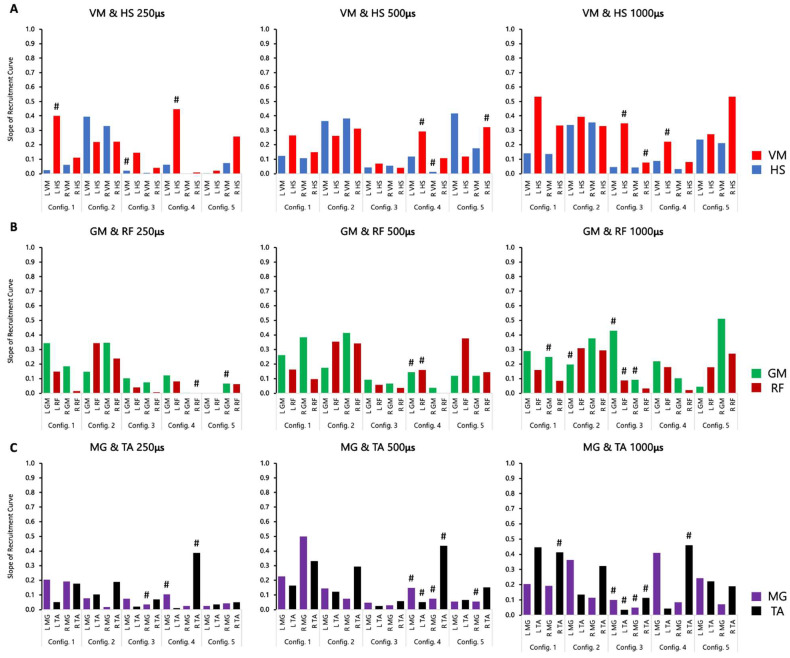
**Individual peak slope values of key extensor and flexor muscle pairs for 0882.** Maximal peak slope of the SCES-evoked recruitment curves was determined for the knee, hip, and ankle extensors and flexors using the five standardized configurations. (**A**) Peak slope for the VM and HS at 250, 500, and 1000 µs. (**B**) Peak slope for the GM and RF at 250, 500, and 1000 µs. (**C**) Peak slope for the GM and RF at 250, 500, and 1000 µs. Config. 1: wide-field configuration with caudal cathodes and rostral anodes; Config. 2: wide-field configuration with rostral cathodes and caudal anodes; Config. 3: central narrow-field configuration with rostral anodes and caudal cathodes; Config. 4: caudal narrow-field configuration with rostral anodes and caudal cathodes; Config. 5: rostral narrow-field configuration with rostral cathodes and caudal anodes. VM: vastus medialis; HS: hamstrings; GM: gluteus medius; RF: rectus femoris; MG: medial gastrocnemius; TA: tibialis anterior. ^#^ Indicates sigmoidal fit had an R^2^ < 0.8.

**Figure 8 jcm-13-01344-f008:**
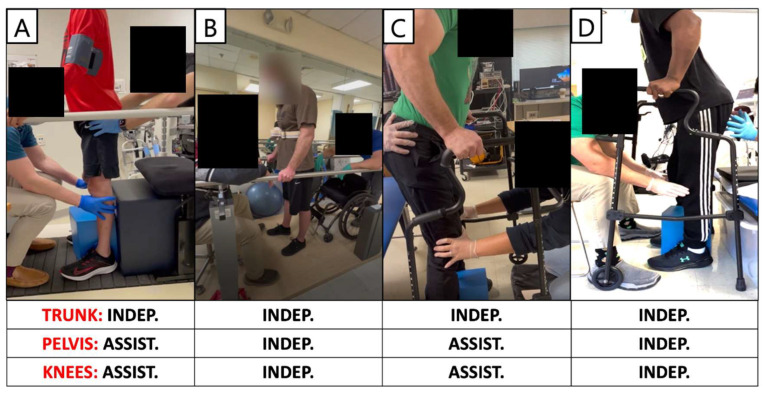
**Standing performance after using pSCES in persons with chronic SCI.** Panels **A**–**D** highlight standing ability assessment for each participant. Independent standing ability was assessed by whether participants required external assistance from study staff at the trunk, pelvis, or knees to maintain standing position using parallel bars or walker. (**A**) Participant 0881 achieved standing without external assistance at the trunk but required external assistance at the pelvis and knees using three pSCES configurations, all with caudal cathodes and rostral anodes delivered simultaneously. (**B**) Participant 0882 achieved overground standing in parallel bars independently with no external assistance from research staff at the trunk, pelvis, and knees using two pSCES configurations both with rostral cathodes and a mix of rostral and caudal anodes delivered simultaneously. (**C**) Participant 0883 was unable to achieve independent overground standing using walker and required external assistance from research staff at the pelvis and knees during sit to stand attempts. Two pSCES configurations both with rostral cathodes and anodes and caudal cathodes and anodes were both delivered simultaneously to assist in standing. (**D**) Participant 0884 was able to achieve standing with external assistance at the pelvis and knees during the sit-to-stand transition using a walker. External assistance was reduced gradually once the participant was in an upright position, and the participant was able to maintain standing with assistance from upper extremities. One pSCES configuration with rostral anodes and caudal cathodes was delivered to enable standing. Indep: Independent; Assist.: Assistance.

**Figure 9 jcm-13-01344-f009:**
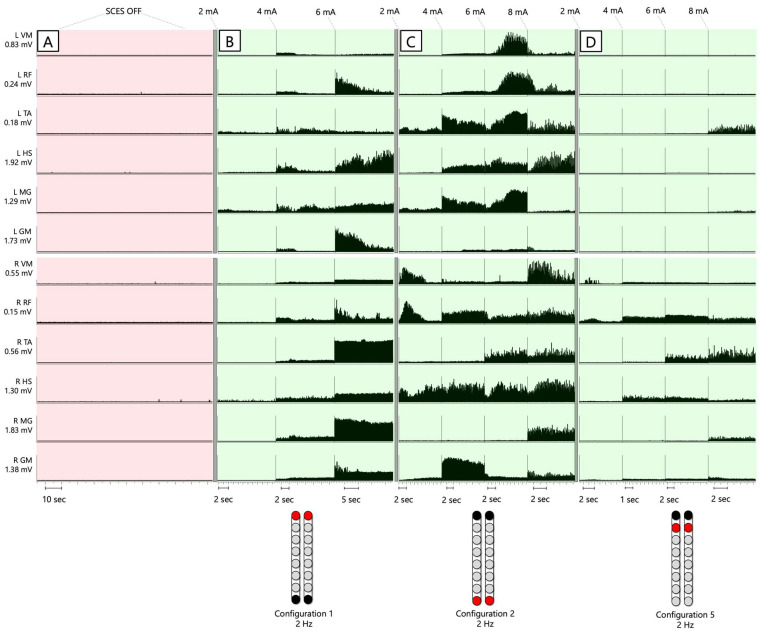
**Representative lower limb EMGs of 0882 in an upright standing position using a standing frame.** Electromyograms (EMGs) of lower extremity muscles of 0882 during standing while three individual standardized SCES configurations were delivered at a pulse duration of 500 µs. (**A**) left and right leg muscles, during upright supported standing with SCES off. (**B**) Left and right leg muscles when the wide-field configuration with caudal cathodes and rostral anodes was delivered. (**C**) Left and right leg muscles when the wide-field configuration with rostral cathodes and caudal anodes was delivered. (**D**) Left and right leg muscles when the rostral narrow-field configuration with rostral cathodes and caudal anodes was delivered. EMGs presented are rectified and band-pass filtered at 10–990 Hz. Pink panel represents pSCES off and green panels represent pSCES on. VM: vastus medialis; RF: rectus femoris; TA: tibialis anterior; HS: hamstrings; MG: medial gastrocnemius; GM: gluteus medius; sec: seconds; Hz: hertz; µs: microseconds.

**Table 1 jcm-13-01344-t001:** Clinical characteristic of individuals with SCI.

ID	Sex	Age (yrs.)	Weight (kg)	Height (cm)	BMI	TSI (yrs.)	NLI	AIS	Classification
0881	M	25	48.6	174.3	16.0	6	C8	A	Tetraplegia
0882	M	36	99.4	182.2	29.9	9	T11	B	Paraplegia
0883	M	38	99.0	180.5	30.4	12	T6	A	Paraplegia
0884	M	54	93.4	182.8	28.0	24	T4	A	Paraplegia

Yrs.: years; kg: kilograms; cm; centimeters; kg/m^2^: kilograms per meter squared; BMI: body mass index; TSI: time since injury; AIS: American Spinal Injury Association Impairment Scale; M: male; NLI: neurologic level of injury.

**Table 2 jcm-13-01344-t002:** Axial MRI biomarker measurements prior enrollment in the trial.

ID	Lesion Length (mm)	Lesion Volume (mm^3^)	Axial Damage Ratio
0881	32.0	1932.3 mm^3^	0.92
0882	23.8	1881.2 mm^3^	0.81
0883	88.1	Unable to Assess	Unable to Assess
0884	15.5	Unable to Assess	0.8

**Table 3 jcm-13-01344-t003:** Peak slope ratio of the recruitment curves following applications of 5 different configurations of pSCES for three pairs of agonist–antagonist muscle groups in persons with chronic SCI.

		**Configuration 1** Wide field configuration 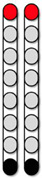						**Configuration 2** Wide field configuration 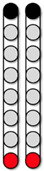						**Configuration 3** Central narrow field configuration 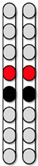						**Configuration 4** Caudal narrow field configuration 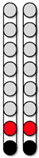						**Configuration 5** rostral narrow field configuration 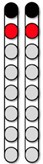					
		Left			Right			Left			Right			Left			Right			Left			Right			Left			Right		
Pulse Duration (µs)		250	500	1000	250	500	1000	250	500	1000	250	500	1000	250	500	1000	250	500	1000	250	500	1000	250	500	1000	250	500	1000	250	500	1000
0881	VM/HS	0.38	1.14 ^+#^	0.33	0.17	0.77 ^#^	0.72	0.46	0.39	0.32	0.24	0.19	0.16	0.52	1.31 ^+#^	0.83 ^#^	0.17	0.76 ^#^	0.34 ^#^	0.26	0.38	0.39	0.24	0.23	0.30	0.90	0.48	0.90	1.91 ^+^	0.52	0.79
	GM/RF	1.59 ^+#^	1.05 ^+#^	1.67 ^+#^	1.76 ^+#^	1.01 ^+#^	1.91 ^+#^	1.55 ^+#^	3.44 ^+#^	8.64 ^+#^	1.86 ^+#^	3.90 ^+#^	2.24 ^+#^	0.96 ^#^	1.17 ^+#^	1.19 ^+#^	1.63 ^+#^	1.13 ^+#^	1.08 ^+#^	8.75 ^+#^	2.32 ^+#^	1.48 ^+#^	0.81 ^#^	2.43 ^+#^	1.18 ^+#^	0.68	0.42	2.12 ^+^	0.67	0.20	0.43
	MG/TA	1.05 ^+#^	0.94	1.11 ^+#^	1.30 ^+#^	0.89	1.15 ^+#^	1.77 ^+#^	1.76 ^+#^	1.04 ^+#^	1.40 ^+#^	1.16 ^+#^	1.25 ^+#^	1.02 ^+^	0.91	0.71	0.48 ^#^	0.93	0.86	1.15 ^+#^	0.93 ^#^	1.06 ^+#^	1.89 ^+#^	1.02 ^+#^	1.06 ^+#^	4.55 ^+^	0.42	0.02	9.04 ^+^	1.83 ^+^	0.88
0882	VM/HS	0.06	0.47	0.26	0.54	0.72	0.41	1.80 ^+#^	1.39 ^+#^	0.86 ^#^	1.49 ^+#^	1.23 ^+#^	1.07 ^+#^	0.13	0.62	0.13	0.13	1.34 ^+^	0.54	0.14	0.41	0.39	0.22	0.11	0.39	0.18	3.50 ^+#^	0.87	0.29	0.55	0.40
	GM/RF	2.31 ^+#^	1.62 ^+#^	1.82 ^+#^	12.74 ^+#^	3.97 ^+#^	2.92 ^+#^	0.43	0.49	0.64 ^#^	1.46 ^+#^	1.21 ^+#^	1.28 ^+#^	2.51 ^+#^	1.62 ^+#^	4.93 ^+#^	12.69 ^+#^	1.81 ^+#^	2.80 ^+#^	1.50 ^+#^	0.90 ^#^	1.22 ^+#^	2.57 ^+^	12.54 ^+#^	4.85 ^+#^	0.76	0.32	0.25	1.08 ^+#^	0.84 ^#^	1.89 ^+#^
	MG/TA	4.08 ^+#^	1.39 ^+#^	0.46 ^#^	1.09 ^+^	1.51 ^+#^	0.46 ^#^	0.76	1.19 ^+#^	2.71 ^+#^	0.10	0.26	0.35	3.73 ^+^	1.95 ^+#^	2.84 ^+#^	0.49	0.53	0.44	9.64 ^+#^	2.93 ^+#^	9.34 ^+#^	0.07	0.17	0.18	0.73 ^#^	0.83	1.09 ^+#^	0.84	0.37 ^#^	0.37 ^#^
0883	VM/HS	3.95 ^+#^	1.82 ^+#^	0.79	0.55 ^#^	0.71 ^#^	0.44	9.24 ^+^	0.94	0.82 ^#^	0.13	1.53 ^+^	0.81 ^#^	0.35	1.69 ^+^	6.95 ^+#^	1.05 ^+^	0.68	0.66	0.14	0.50	2.81 ^+^	5.12 ^+^	0.97 ^#^	0.83 ^#^	23.62 ^+^	1.14 ^+^	0.67	0.44	0.64	0.12
	GM/RF	Malfunction of GM senor																													
	MG/TA	0.10	17.50 ^+^	4.47 ^+^	1.73 ^+^	2.85 ^+^	1.86 ^+^	0.03	0.33	0.80	0.85	3.43 ^+^	0.75	0.30	0.25	0.08	0.67	0.53	0.59	0.08	0.01	0.58	1.64 ^+^	2.61 ^+^	2.06 ^+^	0.14	0.12	0.04	0.25	0.76	0.38
0884	VM/HS	0.45	0.40	0.86 ^#^	0.27	0.54	0.53 ^#^	0.56	*	1.11 ^+^	8.11 ^+^	*	0.04	2.57 ^+^	*	*	4.68 ^+^	*	*	0.04	0.19	0.20	0.08	0.09	0.11	Activating Abdominal Muscles					
	GM/RF	Malfunction of GM senor																													
	MG/TA	1.55 ^+#^	1.23 ^+#^	2.03 ^+#^	1.03 ^+#^	0.80 ^#^	0.88 ^#^	4.77 ^+^	*	0.02	0.08	*	4.06 ^+^	0.17	*	*	0.33	*	*	3.24 ^+^	2.47 ^+#^	0.81 ^#^	1.55 ^+#^	0.68 ^#^	0.90 ^#^	Activating Abdominal Muscles					

VM: vastus medialis; HS: hamstrings; GM: gluteus medius; MG: medial gastrocnemius; TA: tibialis anterior muscles; ^#^ indicates configuration was clinically determined suitable for standing when recruitment curves were visually assessed. ^+^ indicates configuration was determined suitable for standing based on slope ratios. * Indicates not enough data points were tested to yield slope.

**Table 4 jcm-13-01344-t004:** Agreement between the pSECS evoked potential of different muscle groups and the slope ratio of the recruitment curves of 5 different configurations in persons with SCI.

	**Configuration 1** Caudal wide field Configuration	**Configuration 2** Rostral wide field Configuration	**Configuration 3** Central narrow field configuration	**Configuration 4** Caudal narrow field configuration	**Configuration 5** Rostral narrow field configuration	
	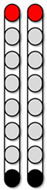	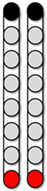	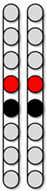	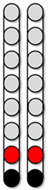	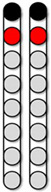	
**Subject ID**		**Selected Configurations Based on the Evoked Potentials**	**Recommended Configurations Based on the Slope Ratio of the Recruitment Curves**	**Agreement**	**Potential Recommendations**	**Standing Performance**
0881	VM/HS	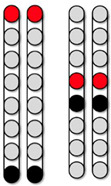 15 Hz & 700–750 µs	Config. 1 and 3 @ 500 µs	Config. 1 and 3	Perfect Agreement	- Trunk control with support at both knees
	GM/RF		Config. 1, 2, 3, 4 @ 250, 500, 1000 µs	Config. 1 and 3	Potentially Test config. 2 or 4	
	MG/TA		Config. 1, 2, 4 @ 250, 500, 1000 µs	Config. 1	Potentially Test config. 2 or 4	
0882	VM/HS	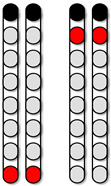 15 Hz & 500 µs	Config. 2,5 @ 500 µs	Config. 2 and 5	Perfect Agreement	- Independent standing between the parallel bars or with a roller walker. - The config. 2 and 5 were common across the agonist-antagonist muscle groups
	GM/RF		Config. 1, 2, 3, 4, 5 @ 250, 500, 1000 µs	Config. 2 and 5	Potentially Test config. 1, 3 or 4	
	MG/TA		Config. 1, 2, 3, 4, 5 @ 250, 500, 1000 µs	Config. 2 and 5	Potentially Test config. 1, 3 or 4	
0883	VM/HS	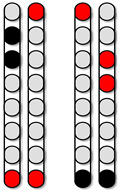 10–15 Hz & 370–760 µs	Config. 1, 3 @ 250–1000 µs	Config. 1	Potentially Test config. 3	- Maximum assistance at the hips and knee joints to achieve standing - Potentially testing config. 3 or 4
	GM/RF		Malfunction of the GM sensor			
	MG/TA		Config. 1, 4 @ 250–1000 µs	Config. 1	Potentially Test config. 4	
0884	VM/HS	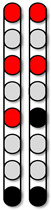 15 Hz & 700 µs	Config. 1, 2 and 3 @ 250–1000 µs	Config. 1	Potentially Test config. 2 and 3	- Independent standing with spot or min ass. to support the knee joints - Potentially test config. 2.
	GM/RF		Malfunction of the GM sensor			
	MG/TA		Config. 1, 2 and 4 @ 250–1000 µs	Config. 1	Potentially Test config. 2 and 4	

## Data Availability

All data reported in this paper will be shared by the lead contact author (Ashraf Gorgey (ashraf.gorgey@va.gov)) upon request.

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
