# Peer review of "Peak Slope Ratio of the Recruitment Curves Compared to Muscle Evoked Potentials to Optimize Standing Configurations with Percutaneous Epidural Stimulation after Spinal Cord Injury"

_jcm, 2024, doi:10.3390/jcm13051344_

Round 1

Reviewer 1 Report

Comments and Suggestions for Authors

Having read the manuscript, I can say that the authors have done a lot of work. Undoubtedly, epidural spinal cord stimulation has become one of the methods for restoring motor function in people with spinal cord injury. It is relevant to develop optimal stimulation protocols for this purpose Since such stimulation was initially used to relieve pain and reduce spastic muscle contraction in patients. Moreover, of course, the strategy of neuromodulation of spinal centers is one of the promising strategies to restore walking in patients receiving SCI.

Title: Regarding the term "Percutaneous Epidural Stimulation", in my opinion it is not correct because Epidural stimulation was performed. This error is throughout the text.

There are specific designations, such as spinal cord stimulation (SCS) and, more specifically, epidural SCS (eSCS), respectively, or percutaneous spinal cord stimulation (pSCS). However, this study is not about percutaneous stimulation, it is about the method of electrode placement.

The methods should describe the implantation procedure in a clearer and more detailed way, possibly with photos, possibly in a supplementary material. Reference [21] does not solve this problem.

Technical errors in the abstract: it should be stated that determined the slope of the recruitment curve of motor units, not people. Thus, line 47, the word "muscles" in "agonist-antagonist groups" is missing. Then, lines 51-52 are about muscles, not people - should be rephrased (something like this: "Two of the four participants advanced an increase in the slope of the recruitment curves of the MUs into standing performance after using tonic stimulation").

Introduction. In terms of meaning the introduction is well written. However, I would like to draw the attention of the authors to the literature used for the analysis. For example, in the first paragraph of lines 59-73, in my opinion, to emphasize what is already known, it is necessary to make references to newer sources, preferably to reviews. Especially since the authors themselves write (lines 62,63): "Over the last decade...".

For example:

Taccola, G., Sayenko, D., Gad, P., Gerasimenko, Y., & Edgerton, V. R. (2018). And yet it moves: Recovery of volitional control after spinal cord injury. Progress in neurobiology, 160, 64–81. https://doi.org/10.1016/j.pneurobio.2017.10.004;

Rejc, E., & Angeli, C. A. (2019). Spinal Cord Epidural Stimulation for Lower Limb Motor Function Recovery in Individuals with Motor Complete Spinal Cord Injury. Physical medicine and rehabilitation clinics of North America, 30(2), 337–354. https://doi.org/10.1016/j.pmr.2018.12.009;

Hachmann, J. T., Yousak, A., Wallner, J. J., Gad, P. N., Edgerton, V. R., & Gorgey, A. S. (2021). Epidural spinal cord stimulation as an intervention for motor recovery after motor complete spinal cord injury. Journal of neurophysiology, 126(6), 1843–1859. https://doi.org/10.1152/jn.00020.2021

Skiadopoulos, A., Pulverenti, T. S., & Knikou, M. (2022). Physiological effects of cathodal electrode configuration for transspinal stimulation in humans. Journal of neurophysiology, 128(6), 1663–1682. https://doi.org/10.1152/jn.00342.2022

D'hondt, N., Marcial, K. M., Mittal, N., Costanzi, M., Hoydonckx, Y., Kumar, P., Englesakis, M. F., Burns, A., & Bhatia, A. (2023). A Scoping Review of Epidural Spinal Cord Stimulation for Improving Motor and Voiding Function Following Spinal Cord Injury. Topics in spinal cord injury rehabilitation, 29(2), 12–30. https://doi.org/10.46292/sci22-00061

Finn, H. T., Bye, E. A., Elphick, T. G., Boswell-Ruys, C. L., Gandevia, S. C., Butler, J. E., & Héroux, M. E. (2023). Transcutaneous spinal stimulation in people with and without spinal cord injury: Effect of electrode placement and trains of stimulation on threshold intensity. Physiological reports, 11(11), e15692. https://doi.org/10.14814/phy2.15692

Verma, N., Romanauski, B., Lam, D., Lujan, L., Blanz, S., Ludwig, K., Lempka, S., Shoffstall, A., Knudson, B., Nishiyama, Y., Hao, J., Park, H. J., Ross, E., Lavrov, I., & Zhang, M. (2023). Characterization and applications of evoked responses during epidural electrical stimulation. Bioelectronic medicine, 9(1), 5. https://doi.org/10.1186/s42234-023-00106-5

Ramadan, A., König, S. D., Zhang, M., Ross, E. K., Herman, A., Netoff, T. I., & Darrow, D. P. (2023). Methods and system for recording human physiological signals from implantable leads during spinal cord stimulation. Frontiers in pain research (Lausanne, Switzerland), 4, 1072786. https://doi.org/10.3389/fpain.2023.107278

I would especially like to note paragraph 2 (lines 74-83) - references only to own publications are given. Therefore, it is necessary to write that in "our research". However, I would like to note that 25% of there are likely many references to own work, there are publications by other groups, for example:

Sharif, S., & Jazaib Ali, M. Y. (2020). Outcome Prediction in Spinal Cord Injury: Myth or Reality. World neurosurgery, 140, 574–590. https://doi.org/10.1016/j.wneu.2020.05.043

Berliner, J. C., O'Dell, D. R., Albin, S. R., Dungan, D., Sevigny, M., Elliott, J. M., Weber, K. A., Abdie, D. R., Anderson, J. S., Rich, A. A., Seib, C. A., Sagan, H. G. S., & Smith, A. C. (2023). The influence of conventional T2 MRI indices in predicting who will walk outside one year after spinal cord injury. The journal of spinal cord medicine, 46(3), 501-507.  https://doi.org/10.1080/ 10790268.2021.1907676

Methods: In my opinion, it is necessary to clearly describe the implantation procedure and the stimulation procedure to clarify what kind of stimulation we are talking about. The level of electrode placement and the method of attachment are not specified. The lack of a study design for understand how long the study lasted, when the evaluation was performed. Were the samples done all at once or once, there was a sequence of changing stimulation modes. How long after the regimen change were the EMG parameters recorded again? Is it the same in all patients?

Technical notes:

Line 273 - the word "muscle" is missing. In Latin names, the word "muscle" is always present, and the names are italicized, for example, m. tibialis anterior. If the word "muscle" is missing in the description, then, at the beginning of the enumeration, it is necessary to write "muscles" of the tibia and thigh and further enumeration.

The terms used require clarification. The term 'motor evoked potential' (MEP) most commonly refers to the action potential elicited by noninvasive stimulation of the motor cortex through the scalp.

North RB, Lempka SF, Guan Y, et al. Glossary of Neurostimulation Terminology: A Collaborative Neuromodulation Foundation, Institute of Neuromodulation, and International Neuromodulation Society Project. Neuromodulation. 2022;25(7):1050-1058. doi:10.1016/j.neurom.2021.10.010

Sivanesan E, North RB, Russo MA, et al. A Definition of Neuromodulation and Classification of Implantable Electrical Modulation for Chronic Pain. Neuromodulation. 2024;27(1):1-12. doi: 10.1016/j.neurom.2023.10.004

See Figure 1. In general, the caption does not match the figure. The description needs to be revised and expanded. What, for example, do your arrows mean? (B) You should enlarge the implantation site and correctly indicate the projection to the vertebrae. (C) Did you correctly mark the muscles that were examined? It is better to indicate with dots the place of EMG withdrawal. (D) the captions within the figure need to be enlarged. Is it necessary to indicate the parameters of stimulation - after all you want to demonstrate the configuration. (E) Here, it is necessary to specify the stimulation parameters, the place of stimulation and configuration, for which muscle, scale. (F) Look carefully; you have mixed up the axis designations and you also need to specify the stimulation conditions and which muscle. What does Y-axis "Recruitment (V/Vmax)»? This parameter is not described in the methods section. The X-axis is not the amplitude but the intensity of stimulation (see the glossary links above). Importantly, when stimulating the spinal cord, the response from muscles of complex configuration. This is why such a parameter the "area under the curve" is usually used. Therefore, it is not clear by which segment of the response you determined the amplitude of the signal to construct the recruitment curve; it may also be worthwhile to cite such a figure in the methods.  For example, look from your list at number [20] Sayenko DG, Atkinson DA, Dy CJ, et al. Spinal segment-specific transcutaneous stimulation differentially shapes activation pattern among motor pools in humans. J Appl Physiol (1985). 2015;118(11):1364-1374. doi:10.1152/japplphysiol.01128.2014

And most importantly, how was the slope of the recruiting curve determined? The whole article is devoted to this and not a word in the methods.

Results

Figure 3 shows that patients B and C (not written in the caption of the figure but to be guessed) strongly displaced the electrodes. Given that the site of injury was also not similarly (C8 to Th11), the differences were very large. This is important in terms of obtaining optimal responses. See: your number [39] Sayenko DG, Angeli C, Harkema SJ, Edgerton VR, Gerasimenko YP. Neuromodulation of evoked muscle potentials induced by epidural spinal-cord stimulation in paralyzed individuals. J Neurophysiol. 2014 Mar;111(5):1088-99. doi: 10.1152/jn.00489.2013.

Figure 3 itself requires deciphering the signatures. And, probably, it is necessary to indicate the designations related to the patients from Table 1.

Question: why is the number of configurations different for different patients? It becomes clear only after reaching 3.5. We should move this figure to 3.5. Then, figures 4-6 - it is not clear how they were constructed. What is taken as 100%. There is no explanation in the figure caption. What does green and gray background mean? This is for drawings in the supplementary material. Note the error in the stimulus duration dimension label.

Table 3 needs to be made readable.

Figure 7 you should also add a description, write the units of measurement on the axes.

Figure 8 - The title of the figure does not correspond to the image. Add a description of the figure, not only in the caption but also probably in the text.

Perhaps Figure 3 should be moved to this part (subchapter 3.5), since the references are always going to it. The information in the figure corresponds more to subchapter 3.5.

In general, I would like to say that the results are described carelessly. It is very difficult to read. Different stimulation frequencies and durations were not mentioned in the methods or results section. There is a need for clarity. Were the optimal modes chosen based on the slope of the recruitment curves or on the activity of the extensor muscles? Then, the hypothesis should be changed in the introduction and the title of the article should be different.

Discussion

The sources involved correspond to the latest understanding of the problem of spinal cord stimulation. However, as mentioned above, 25% of the sources are the authors' own research. I think that today there are enough groups that are engaged in the development of the presented issue and can be quoted in the present manuscript.

The discussion about the peak set of the recruitment curve is not quite clear. Since the slope of the curve can only be understood by reaching the maximum amplitude. In principle, the slope is analyzed to understand how the recruitment of large and small MUs has changed. If the excitability of small neurons changes; sometimes, the set of curves splits into two plateaus, and then, there will be two slopes. In addition, depending on the location of stimulation, both the slope and the reaching the plateau of the MU recruitment curve will also change. Since it is difficult to understand from the text what you mean by the slope of the curve, it is difficult to determine what, in fact, you were aiming at. Since the electrodes were placed very differently in the patients, knowing that the anatomical location of the MN pools is different in the thigh and tibia muscles, again, is there no error? Judging by the results, the optimal strategy was to look at the contraction of the extensor muscles.

Author Response

Point-by-point response to reviewer 1 is attached. 

Reviewer 2 Report

Comments and Suggestions for Authors

I have reviewed the manuscript by Alazzam et al., titled "The Use of Peak Slope Ratio to Identify Standing Configurations following Percutaneous Epidural Stimulation in Persons with Spinal Cord Injury." The study investigates the effects of wide and narrow-field configurations of percutaneous spinal cord epidural stimulation (pSCES) on motor recruitment curves in individuals with motor complete spinal cord injury (SCI).

The manuscript provides a clear overview of the study, emphasizing the shift from relying solely on motor evoked potentials (MEPs) to exploring the impact of different pSCES configurations on motor recruitment curves. The use of magnetic resonance imaging (MRI) to individualize participants based on lesion characteristics is a commendable approach.

Consider adding a concept diagram to visually represent the key findings and the methodology. This could aid readers in quickly grasping the main take-home message.   Some sentences could be revised for improved clarity and flow. For example, consider rephrasing: "The findings suggest that using the peak slope ratio demonstrates potential to identify the pSCES configurations necessary to achieve standing in persons with SCI." and a few others in the manuscript.   I recommend minor revisions to enhance clarity and consider the addition of visual elements to improve the manuscript's overall quality.    

Comments on the Quality of English Language

OK

Author Response

Point-by-point response to reviewer is attached. 

Round 2

Reviewer 1 Report

Comments and Suggestions for Authors Dear authors! You've done a great job. Thank you very much for your research. There are only a few technical comments: 1. Authors added the word “muscle” to the methods and that’s enough. It’s no need to use the letter "m." throughout the whole text. 2. Please check units of signal duration in all Figures, especially in Figures 4-6. The letter "μ" is not readable in the Figures. 3. The numbering of subheadings in the methods is incorrect.   All in all, best wishes for continued researches

Author Response

Dear authors! You've done a great job. Thank you very much for your research. There are only a few technical comments

Thank you so much for your time and effort reviewing our work.

  1. Authors added the word “muscle” to the methods and that’s enough. It’s no need to use the letter "m." throughout the whole text.

Thank you.  The letter m. was deleted from the entire text.

  1. Please check units of signal duration in all Figures, especially in Figures 4-6. The letter "μ" is not readable in the Figures.

We totally agree and the letter u was erroneously used instead of the correct symbol “μ. We have replaced figures 4-6 and all the other supplementary files to correct for this problem.

  1. The numbering of subheadings in the methods is incorrect.

Thank you. This was fixed as requested.